# Giant Fern Genomes Show Complex Evolution Patterns: A Comparative Analysis in Two Species of *Tmesipteris* (Psilotaceae)

**DOI:** 10.3390/ijms24032708

**Published:** 2023-01-31

**Authors:** Pol Fernández, Ilia J. Leitch, Andrew R. Leitch, Oriane Hidalgo, Maarten J. M. Christenhusz, Lisa Pokorny, Jaume Pellicer

**Affiliations:** 1Institut Botànic de Barcelona (IBB, CSIC-Ajuntament de Barcelona), Passeig del Migdia s.n., Parc de Montjuïc, 08038 Barcelona, Spain; 2Facultat de Farmacia i Ciències de l’alimentació, Campus Diagonal, Universitat de Barcelona, Av. de Joan XXIII, 27-31, 08028 Barcelona, Spain; 3Royal Botanic Gardens, Kew, Richmond TW9 3AE, UK; 4School of Biological and Behavioural Sciences, Queen Mary University of London, London E1 4NS, UK; 5Real Jardín Botánico (RJB-CSIC), Plaza de Murillo 2, 28014 Madrid, Spain

**Keywords:** genome size, polyploidy, monilophytes, pteridophytes, repetitive DNA, transposable elements

## Abstract

Giant genomes are rare across the plant kingdom and their study has focused almost exclusively on angiosperms and gymnosperms. The scarce genetic data that are available for ferns, however, indicate differences in their genome organization and a lower dynamism compared to other plant groups. *Tmesipteris* is a small genus of mainly epiphytic ferns that occur in Oceania and several Pacific Islands. So far, only two species with giant genomes have been reported in the genus, *T. tannensis* (1C = 73.19 Gbp) and *T. obliqua* (1C = 147.29 Gbp). Low-coverage genome skimming sequence data were generated in these two species and analyzed using the RepeatExplorer2 pipeline to identify and quantify the repetitive DNA fraction of these genomes. We found that both species share a similar genomic composition, with high repeat diversity compared to taxa with small (1C < 10 Gbp) genomes. We also found that, in general, characterized repetitive elements have relatively high heterogeneity scores, indicating ancient diverging evolutionary trajectories. Our results suggest that a whole genome multiplication event, accumulation of repetitive elements, and recent activation of those repeats have all played a role in shaping these genomes. It will be informative to compare these data in the future with data from the giant genome of the angiosperm *Paris japonica*, to determine if the structures observed here are an emergent property of massive genomic inflation or derived from lineage specific processes.

## 1. Introduction

Fern genome dynamics have not been studied as much as in other plant groups, especially when compared with angiosperms and to a lesser extent gymnosperms [1,2], although efforts are being made to progress our understanding of their genome evolution and organization (e.g., [3,4,5]). Some of the early-diverged lineages (e.g., Osmundaceae) have shown relatively limited genome size (GS) range, chromosome diversity and low sequence substitution rates [6]. Indeed, the discovery of an extremely well preserved fossil featuring chromosomes and nuclei, showing clear affinities to extant relatives, led the authors to hypothesize that fern genomes have remained remarkably stable over time [7].

Early studies exploring the genomic landscape of ferns suggested that repetitive elements did not play a major role in driving fern genome dynamics [8]. Instead, whole genome multiplications (WGMs) were suggested as the major diversification driver of both GS and chromosome evolution in the group [9]. Intriguingly, fern genomes are relatively large (mean GS, 1C = 14.32 Gbp) compared with other plant groups, notably, angiosperms (mean 1C = 5.02 Gbp), lycophytes (mean 1C = 1.63 Gbp), and bryophytes (mean 1C = 0.57 Gbp), but not when compared with gymnosperms (mean 1C = 18.23 Gbp) [10]. However, large genomes in ferns are frequently linked with WGMs, but they are not necessarily with gymnosperms. Indeed, the combined effect of a relatively high incidence of WGMs in ferns, as hypothesized by large chromosome numbers, is thought to have played a significant role in shaping GS in the majority of ferns, although with exceptions (e.g., *Adiantum capillus-veneris* L. (Pteridaceae); [11]). Certainly, post-polyploidy diploidization restructuring, that is considered to impact most angiosperm genomes [12,13], is not so apparent in polyploid fern genomes, which tend to retain most of the duplicated DNA, resulting in higher proportions of repetitive DNA and WGM-derived genes (e.g., [14,15]).

The continuous and rapid advances in high throughput sequencing (HTS) technologies are enhancing our ability to gain key insights into how plant genomes are organized and function, and are challenging previously accepted views on fern genome dynamics. For example, changes in repetitive DNA landscape of the genome can now be traced and investigated using these technological and analytical developments, such as the RepeatExplorer pipeline [16], which enables the identification and quantification of repetitive elements from short-read sequencing data without the need of reference genomes. In addition, in recent studies, long- and short-read whole genome sequencing (WGS) data of *Ceratopteris richardii* Brongn. (Pteridaceae; i.e., the so-called C-fern genome) have identified a significant accumulation and activation of transposable elements (TE) following WGMs [4,17]. This C-fern genome and the chromosome level assembly of *A. capillus-veneris* by Fan et al. [11] provide the first evidence of TE accumulation in ferns, which is a well-known phenomenon in other plant groups, such as angiosperms [18]. Transposable element accumulation and turnover is well described in angiosperms, and is thought to underpin much of GS variation between species [10,19]. A few previous studies focusing on the repetitive element composition in fern genomes indicate that patterns may be broadly similar to those of other land plants studied to date [3]. However, very little is known about how TE dynamics impacts fern GS and especially amongst the largest fern genomes, the purpose of this study [9,14].

The genus *Tmesipteris* Bernh. is one of the two extant genera in family Psilotaceae J.W. Griff. & Henfr. (also known as whisk-ferns) and it consists of about 15 species (see [20,21] for a review). They are found across the Southern Pacific, Australia, and New Zealand, where they live mostly as epiphytes on tree fern species of the genera *Alsophila* R.Br., *Cyathea* Sm. and *Dicksonia* L’Hér. (Cyatheaceae), with few species such as *Tmesipteris vieillardii* P.A. Dang, growing terrestrially on ultramafic rocks. In this study, we focus on two species, namely *Tmesipteris tannensis* (Spreng.) Bernh. and *T. obliqua* Chinnock, which are found in New Zealand and Eastern Australia, respectively. Both species have giant genomes but differ in ploidy level. *Tmesipteris tannensis* is considered to be a tetraploid, with 2n = 208 chromosomes [22] and a GS of 1C = 73.19 Gbp, while *T. obliqua* is considered to be octoploid, with 2n = 416 and 1C = 147.29 Gbp [9,23]. The latter is, indeed, the largest genome so far reported for any fern, and is comparable in size to the largest known eukaryotic genome, in the angiosperm *Paris japonica* Franch. (Melanthiaceae, 1C = 148.88 Gbp [24]). The two species are of considerable interest because they combine polyploidy and genomic gigantism to a level that has never been addressed before in any group.

Giant genomes (i.e., 1C > 35 Gbp) have evolved in angiosperms and gymnosperms through a variety of pathways, including the amplification of repetitive elements (mainly TEs), which combined with low rates of DNA elimination, lead to the accumulation of DNA and hence GS expansion over evolutionary time [25]. Despite ongoing advances in genomics research, which make ambitious sequencing approaches more feasible than ever before, most studies on ferns have focused on relatively small fern genomes (1C < 15 Gbp, [4,11,14,15,26]), including the filmy fern *Trichomanes speciosum* Willd. (Syn. *Vandenboschia speciosa* (Willd.) G. Kunkel; 1C = 10.5 Gbp [3]) and the C-fern, *C. richardii* (1C = 9.68–11.25 Gbp [4,17]). The later species has become a model plant in fern genomics and developmental studies [26]. Therefore, the study of the giant genomes of *Tmesipteris* here contributes much needed insight into the structure and evolution of giant genomes and expands our meagre understanding on fern genomes [27].

Based on the above, we have carried out a comparative study using next generation sequencing short-read data to explore these genomes with the following main objectives: (i) to identify and quantify the main types of DNA repeats shaping giant fern genomes; (ii) to study the dynamics of repeat turnover over time to see whether repetitive DNA sequences have accumulated gradually over evolutionary history or show bursts of amplification (including recent TE activation), and (iii) to understand the impact of polyploidy on the genome of *T. obliqua,* through a comparative analysis with the genome of *T. tannensis*, in which a duplication of the nuclear DNA content is observed.

## 2. Results

### 2.1. Repeat Composition in Tmesipteris tannensis and Tmesipteris obliqua

Both species share a similar composition of repetitive elements, which is provided in Table 1 and illustrated in Figure 1. The genomic coverages for each species and information on the number of reads analyzed individually can be found in Appendix A. The largest cluster of repetitive DNA found comprised 2.2 and 2.4% of the genome and the seven to eight most abundant clusters made up broadly 10% of the genome in *T. obliqua* and *T. tannensis* respectively. The most abundant repetitive elements belonged to the Ty1/Copia superfamily (16.45% *T. obliqua* and 14.13% *T. tannensis*), followed by Ty3/Gypsy (9.07% *T. obliqua* and 9.36% *T. tannensis*), and satellite DNA (8.04% and 6.47%, respectively). Most Ty1/Copia-like elements were subclassified as Tork and Ivana lineages (see Table 1 for detailed classification). Amongst Ty3/Gypsy-like elements, Athila elements dominate within the superfamily (88.3% *T. obliqua* and 93.9% *T. tannensis* out of Ty3/Gypsy-like elements), while Tekay and Reina elements comprise the rest. The only non-LTR retrotransposons recovered in both species were LINE elements. A total of 32.46% of the genome of *T. obliqua* and 28.92% of the genome in *T. tannensis* (including those repeats with a genomic proportion < 0.01%), do not match any elements in the REXdb database). Comparative analyses of the repeat clusters (Appendix A) revealed that most clusters contain reads from both species (see Table 1 for non-shared repetitive element types).

The fraction of the genome that was identified as single and low copy repeats (GP < 0.01% and not resolved by Repeatexplorer2) mostly comprises the ‘dark matter’ (defined by the sequences that could not be annotated) of the genome [28,29]. This fraction represents a higher fraction of the genome in *T. tannensis* (30.73%) than *T. obliqua* (22.74%). In contrast, the genome of *T. obliqua* has a higher proportion of repetitive elements including almost all individual TE families and tandem repeats. Only Ty3/Gypsy-like elements maintain almost identical proportions in both species. Differences in GS between the two species are explained well in a regression analysis of GS against read numbers in RepeatExplorer2 Ty1/Copia and Ty3/Gypsy repeat clusters (Figure 2, R^2^ > 0.9). The slope of the regression analysis indicated that Ty1/Copia elements were moderately over-represented in *T. obliqua* (slope > 2.01), while Ty3/Gypsy elements were slightly over-represented in *T. tannensis* (slope < 2.01). The regression considering all clusters together explained less of the data (R^2^ = 0.605) than when unclassified clusters were excluded from the analysis (R^2^ = 0.886; see Table 2 for complete statistics).

### 2.2. Heterogeneity among Copies of Repetitive Elements

The analysis of sequence heterogeneity between different reads within repeat clusters that were generated from the comparative analysis (see Methods) showed differences between the two species. Most analyzed clusters for different repeat types in *T. tannensis* showed a skewed distribution towards possessing less conserved sequences (i.e., <94% similarity, Figure 3; see more in Appendix A). By contrast, in *T. obliqua*, we observed a mixture of patterns, some elements (e.g., Ty1/Copia Ivana Cluster 15, Figure 3(C2)) showed the same type of distribution as in *T. tannensis,* while others (e.g., the TIR/CACTA repeat in Cluster 76 (Figure 3(F2)) and the LINE element in Cluster 41 (Figure 3(E2))) presented a less skewed distribution indicating more homogeneous sequences of these repetitive elements (Figure 3). It is worth noting that even clusters from the same element type, sometimes showed different levels of heterogeneity depending not only on the species, but also on the coding domain that was analyzed. For example, the analysis of two clusters comprising Ty1/Copia-Tork element sequences showed that Cluster 18 (comprising the protein coding RT and RH domains) were more conserved than Cluster 7 (comprising the protein coding INT and PROT domains). We also took a closer look at the differences between clusters that are more abundant in one of the species vs. the other. We found the number of reads of an element in the genome did not affect the sequence heterogeneity of that element.

### 2.3. Average Insertion Times of LTR-Retrotransposons

LTRharvest found 287 and 315 pairs of LTRs in *T. tannensis* and *T. obliqua*, respectively. All the results presented below come from the analysis with the CD-HIT cleaning step, since the results from both approaches (see Methods) were consistent. Long terminal repeat insertion times ranged from 0 to over 17.4 Mya in both species, as illustrated in Figure 4. The oldest LTR insertion time for *T. obliqua* was 17.42 Mya while for *T. tannensis* was 17.45 Mya. The average insertion time for the LTR pairs was 6.45 ± 3.7 Mya in *T. obliqua* and slightly older, 6.64 ± 3.73 Mya in *T. tannensis*.

## 3. Discussion

This is the first study to explore the repetitive landscape in ferns with giant genomes, hence representing the first attempts to unveil the likely processes driving giant genome evolution in *Tmesipteris* and other ferns. Indeed the genomes of both species of *Tmesipteris* are large for land plants as a whole (range 0.061–148.88 Gbp, mean 1C = 5.50 Gbp), and specifically, for ferns (monilophytes: range 0.74–147.29 Gbp, mean 1C = 14.34 Gbp) [10,30].

The extent to which WGMs have impacted the evolution of *Tmesipteris* is not fully understood, especially because evolutionary relationships between species in the genus are currently unknown. However, based on phylotranscriptomics, a WGM has been inferred to pre-date the divergence of *Psilotum* Sw. (Psilotaceae) and *Tmesipteris*, and a second round of genome doubling has been inferred in the most recent common ancestor of *Tmesipteris* [31,32,33], the latter further supported by the lack of diploid chromosome number reports in the genus. It is thought that post-polyploidization mechanisms impact differentially across land plant lineages [34]. In ferns, WGMs can be accompanied by gene silencing (e.g., [9]) but apparently not by large scale chromosome restructuring and genome downsizing, as occurs in angiosperms [35]. The absence of chromosomal restructuring may explain an almost linear relationship between GS and chromosome number in ferns [36]. Thus, the genome dynamics that are reported here are likely to have been impacted by ancient WGM events shared by both species, and the more recent WGM in *T. obliqua* that resulted in the near doubling of its GS compared with *T. tannensis*. Additionally, these WGM events are also likely to have triggered other processes leading to repeat amplification and repeat deletion, whose frequencies influence repeat abundance and sequence heterogeneity, as well as the quantity of dark matter.

### 3.1. TEs Largely Contribute to the Giant Genomes of Tmesipteris

Both *Tmesipteris* species share a similar population of repetitive elements, with limited numbers of unique TEs (Table 1). The superfamily lineage LINE is in huge abundance compared to other plant species occupying 4.04% and 4.9% of the huge genomes of *T. obliqua* and *T. tannensis*, respectively. However, the superfamily lineages Ty1/Copia and Ty3/Gypsy superfamily lineages make up the bulk of repetitive elements in *Tmesipteris,* as in other plants, including ferns [14]. Of these, Ty1/Copia-like elements dominate the repetitive landscape in *Tmesipteris,* and were found in higher proportion in the larger genome of *T. obliqua* compared with *T. tannensis* (see Figure 2). Similarly, other elements such as DNA transposons and satellites were also present in higher proportions in the larger genome, making the overall proportion of repetitive elements larger (by ~14%) in *T. obliqua*. Elements other than Ty1/Copia and Ty3/Gyspy, collectively made significant contributions to the overall genome composition in both species (approximately 17% of the genome). Overall, there is a diverse composition of repetitive sequences in *Tmesipteris*. In fact, the total repeat fraction is larger in *Tmesipteris* than in previously analyzed fern taxa with smaller genomes [14,17].

### 3.2. The Occurrence of Dark Matter in the Giant Genomes of Tmesipteris

The fraction of the *Tmesipteris* genome that was not identified into repeats by RepeatExplorer2 (30.73% of the genome in *T. tannensis* and 22.74% of the genome in *T. obliqua*) likely represents dark matter. This fraction comprises single copy and low-copy repeat fractions of the genome, as already pointed out by Maumus and Quensville [29]. Recently, Nóvak et al. [2] argued that there is a fundamental shift in repeat dynamics in angiosperm and gymnosperm species with GS above about 10 Gbp/1C, such that the rate of accumulation of repeats in giant genomes greatly exceeds their elimination. The consequence is that repeats that have been amplified typically remain in the genome, where they mutate until finally they are so degraded that they are no longer recognised as repeats. Instead, they occur as low/single copy sequences that lack targets for recombination-based DNA removal, and essentially become fossilized. Given the GS of both *Tmesipteris* species and the high abundance of dark matter, it is likely that similar processes are occurring in *Tmesipteris* as is observed in other plants, including gymnosperms and angiosperms with large genomes [2,25,37].

It is of interest that the proportion of dark matter sequences that were recovered (Table 1) occurs in a higher proportion in the smaller genome of *T. tannensis* (30.73%) than the larger genome of *T. obliqua* (22.74%). This seems in contrast to the work of Nóvak et al. [2], where the proportion of dark matter increases with GS. However, it is worth noticing that their work was all based in the context of diploid genomes. Here in *Tmesipteris*, the predominant cause of GS expansion leading to the giant *T. obliqua* genome is WGM, a process that appears to have changed the dynamic in the repeat/dark matter landscape, by triggering repeat amplification. Nevertheless, the hypothesized loss of c. 11.4 Mb in single copy DNA sequences in *T. obliqua* could have been influenced by chromosome rearrangements or by fractionation of the genome subsequent to WGM [38,39,40], or simply by differences in the GS of *T. tannensis* compared with the true ancestor of *T. obliqua*.

### 3.3. Evolutionary Mechanisms Underpinning Repeat Dynamics in the Evolution of the Giant Tmesipteris Genomes

There are three non-exclusive hypotheses that have been proposed to explain the expansion of genomes to gigantic scales in land plants: (i) WGMs, (ii) accumulation of repetitive DNA sequences due to deficient removal mechanisms and (iii) repeat element activation (i.e., after WGM) [17,19,41]. Based on the results that are presented here, the genus *Tmesipteris* presents evidence that all three mechanisms have been operating. Considering the difference in ploidy level between the two species, a focus on the smaller genome of *T. tannensis* provides an opportunity to study long-term repeat dynamics in a genome that has not been influenced by recent WGM. The heterogeneous pattern of repeats that were observed for this species likely arose from their accumulation and mutation over time. These results are similar to those observed in the giant genome of species in the angiosperm genus *Fritillaria* [25]. It is also notable that in some of the TEs in *Tmesipteris*, there is considerable sequence heterogeneity even in the theoretically most conserved region of TE repeats (i.e., coding regions involved in their amplification and insertion into new regions of the genome). Despite this, it is apparent that there must also have been a slow accumulation of TEs over time, as previously described in studies of other fern genomes [17].

Despite this overall heterogeneity in the repetitive landscape in the tetraploid genome of *T. tannensis*, we identified more conserved TEs, with less heterogeneity in the large octoploid genome of *T. obliqua,* for almost all TE elements analyzed. This supports a scenario in which there was an activation of several repeated families after WGM. Certainly, post-WGM repeat activation is a well-known phenomenon and has been described in numerous studies analyzing many types of repetitive elements [19]. GS has been positively correlated with median insertion times in ferns [42] and to provide further support for post WGM amplification of repeats in the lineage leading to *T. obliqua*, we estimated the average LTR-insertion times for both species. We found little difference in insertion times between the two species. However, the results need to be interpreted with caution, because the analysis was based on a genome skimming approach, as conducted by others (e.g., *T. speciosum* [3]), and not on mapping long read sequences with high genome coverage to genome assemblies, as recommended in Baniaga et al. [42]. The ultimate reason for caution is that random sampling sequencing efforts could be biased towards representing more recently derived and conserved repeats. Nevertheless, the average insertion times that were estimated in *Tmesipteris* (approximately 3–10 Mya, Figure 4) falls within the range that was obtained in fern species with smaller genomes, such as, e.g., *C. richardii* (1C = 11.2 Gbp) and *Pteridium aquilinum* (L.) Kuhn (Dennstaedtiaceae; 1C = 15.6 Gbp) [17,42]. Should the insertion times reported here hold true, then it would imply that there are no great differences in TE insertion rates based on GS or WGM. Thus, to explain the accumulation of DNA leading to the evolution of giant genomes in *Tmesipteris*, and the expansion of repeats following a recent WGM in *T. obliqua*, our results are best interpreted as arising from reduced levels of repeat removal, rather than elevated frequencies of repeat amplification.

## 4. Materials and Methods

### 4.1. Plant Material and DNA Sequencing

The same samples of *T. obliqua* and *T. tannensis* that were used to estimate C-values reported by Clark et al. [9] and Hidalgo et al. [23] have been analyzed in the present study. Details on the provenances of these samples can be found in the original publications. Genomic DNA extractions were carried out following the CTAB method with minor modifications [43] and subsequently purified with NucleoSpin column clean-up following the manufacturer’s protocol: (i) DNA products were run on an agarose 1% gel and quality control assessed using a Qubit 3 fluorometer (Thermo Fisher Scientific, Waltham, MA, USA). (ii) Genomic libraries were prepared using a NEBNext^®^ UltraTM II DNA Library Prep Kit for Illumina^®^ (New England Biolabs, Ipswich, MA, USA) with an average insert size of 350–500 bp and (iii) sequenced on a MiSeq platform with v3 chemistry (Illumina, San Diego, CA, USA), generating 150 nt paired-end reads (0.1× genome coverage) at the Queen Mary University of London Genome Centre.

### 4.2. Graph-Based Clustering in RepeatExplorer2

The quality of forward and reverse raw reads was assessed using FASTQC v0.11.9 (https://www.bioinformatics.babraham.ac.uk/projects/fastqc/; accessed on 15 February 2021). Next, the reads were paired, adapter cleaned, and quality trimmed to 100 bp using Trimmomatic v0.39 [44] with the following settings: AVGQUAL:20 MINLEN:100 LEADING:20 TRAILING:20 SLIDINGWINDOW:4:20. Quality filtered reads were then mapped to a full *Psilotum nudum* (L.) P.Beauv. (Psilotaceae) chloroplast sequence (GenBank acc.: KC117179), and those which mapped were removed from the original dataset for downstream analyses. The remaining nuclear reads were then used as input for the RepeatExplorer2 pipeline [16,28], which builds sequence similarity-based graph clusters (i.e., higher level assembly of read contigs) without the need of a reference genome. A BLAST analysis then matches the reads from each cluster to a database of Viridiplantae repetitive elements to characterize them (REXdb, [45]). The annotation of clusters was made using first the automatic annotations done by RepeatExplorer2 followed by a manual revision of each cluster using a 5% sequence similarity threshold. In some cases, individual clustering was required to assist with the manual annotation.

Using this pipeline, preliminary analyses were run with the original sequencing datasets, one for each species, to estimate the maximum number of reads the pipeline could handle and analyze. After these first pilot tests, two individual analyses with the maximum possible number of reads were run with standard clustering parameters (90% similarity over 55% of the read length, and cluster size threshold = 0.01%). For the individual analysis, we included the same number of reads (i.e., 4 million) for each species and not the same proportion according to their GS to avoid the possible bias due to the amount of data that were analyzed, as suggested by the software developers (*T. tannensis* 0.54% coverage; *T. obliqua* 0.27% coverage). We also performed individual analyses with a proportional number of reads and found consistent results with those that we present here. Also, a comparative analysis was done using the same proportion of reads (i.e., 0.14% of GS) for each species (i.e., *T. tannensis*: 1 million reads; *T. obliqua:* 2 million reads) and annotating clusters using the same parameters as for the individual analysis.

After annotation and quantification of clusters in the individual analysis, a bar plot was constructed to illustrate the composition of each analyzed genome using library ggplot2 [46] in R v4.2.2 [47]. The similarity between the repetitive landscape of both species was evaluated based on the annotated clusters found in the RepeatExplorer2 comparative analysis. Proportions (with respect to GS) and ratios of each cluster were calculated in R. Additionally, a pairwise scatterplot of all clusters annotated with RepeatExplorer2 was constructed to compare the number of reads in each species using ggplot2. From these scatterplots, regressions for the largest repeat families (i.e., LTR retrotransposons Ty1/Copia and Ty3/Gypsy) were calculated using the lm function in R, forcing them to pass through point (0,0). Finally, the slope was compared with the expected value of 2.01, which represents the GS ratio between both species, as in [18].

### 4.3. Repeat Heterogeneity Analysis and Average LTR Insertion Times

To assess the level of conservation between copies of the different types of repetitive elements, we selected the two largest RepeatExplorer2 clusters from each of the top eight most abundant repetitive element types. The largest clusters of repeat element types were selected based on the percentage of hits (combined between both species) in the database of the comparative analysis (except for DNA satellites, where the two largest clusters of satellites from each species were selected). Next, for each cluster, the BLAST results file from the comparative RepeatExplorer2 output was selected, which includes the matches between reads with more than 90% identity over 55% of the length. From those, two datasets were created for each of the clusters: one containing only matches between reads from *T. tannensis* and the other comprising only matches between reads from *T. obliqua*. From each dataset, a histogram depicting the frequencies of paired reads with particular sequence identities was drawn using R resulting in two graphs per cluster, one for each species. Since this analysis was based on Kelly et al. [25], who used only analyzed single-end reads, we also tested if there was any bias when using paired-end reads by performing a pilot test using forward reads only, but no bias was detected.

To assess the average insertion times of TEs, pairs of long terminal repeats (LTRs) were extracted from the contigs that were produced from the individual analyses by RepeatExplorer2 using LTRharvest [31]. The protocol that was used was similar to that described in https://github.com/SIWLab/Lab_Info/wiki/Ageing-LTR-insertions#full-pipeline (accessed on 26 September 2022), with the alignments performed using MUSCLE v3 [48]. To calculate the insertion time, the Kimura’s two-parameter distance [49] between the LTR pairs was calculated using fastdist function from the package fastphylo [50]. The T = K/2μ formula was used to calculate the insertion time, where T = time, K = evolutionary distance per site, and μ = synonymous substitution rates per site per year, estimated to be μ = 4.79 × 10^−9^ which corresponds to that calculated for other fern species [51]. Next, a histogram was constructed using ggplot2 in R to show the distribution of insertion times for each species. Finally, the median and the standard deviation was calculated in R following the reasoning in Baniaga et al. [42].

A second analysis was performed including a cleaning step with CD-HIT [52] to remove contigs that were almost identical and were potentially duplicates. We are aware that rather than removing duplicates, we might be removing variants when applying this filtering step, so we did both analyses to compare the results. Additionally, we repeated the analysis (i) using only the contigs that the RepeatExplorer2 algorithm classified as Ty3/Gypsy, and (ii) with just the Ty1/Copia elements. We did both analyses both with and without the CD-HIT cleaning step.

## 5. Conclusions

This study presents the first insights into the composition of repetitive elements in the giant genomes of two species of *Tmesipteris*, providing evidence of how WGM, repeat amplification, and reduced rates of repeat elimination were the most likely mechanisms involved in the rise of giant genomes in this group of ferns. It also contributes to further our understanding on genome evolution in general, and ferns in particular, indicating that giant genomes are not as static as previously assumed. Future studies should consider expanding the sampling effort across the genus as well as including its sister genus *Psilotum* in which diploid populations or taxa with giant genomes (e.g., c. 70 Gbp/1C) are known. Also, efforts should be made to build a robust phylogenetic framework in this group so that more comprehensive comparative phylogenetic approaches can be applied to better understand TEs evolution across the diversity of extant ploidy levels.

## Figures and Tables

**Figure 1 ijms-24-02708-f001:**
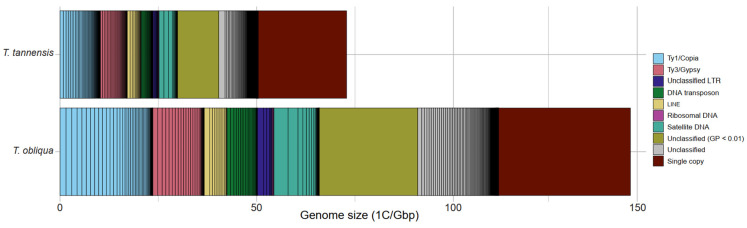
Genomic repeat composition of *T. tannensis* and *T. obliqua*. Genomic proportions (GP) in Gbp/1C are colored by repeat class. The vertical black lines within each repeat class indicate the genomic abundances of particular variants of repeats, as identified from individual RepeatExploer2 clusters. ‘Unclassified’ refers to those repeats that are unidentified with a genomic proportion ≥ 0.01%, while ‘Unclassified (GP < 0.01)’ refers to reads that are part of repeated elements with a genomic proportion below 0.01%.

**Figure 2 ijms-24-02708-f002:**
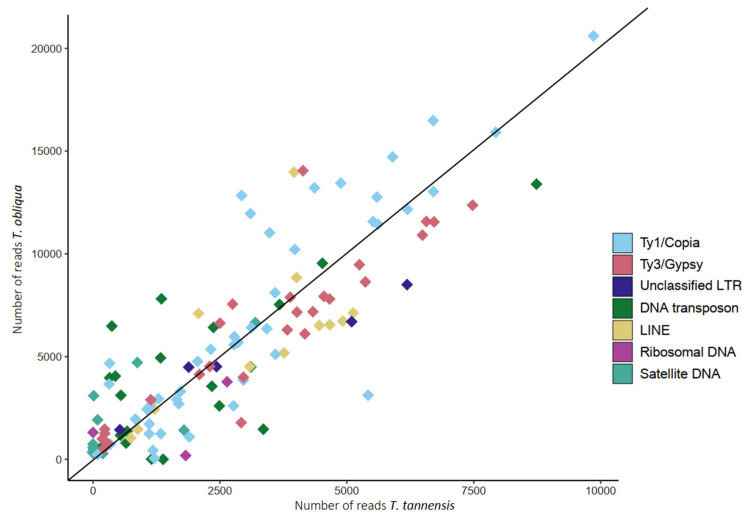
Pairwise scatterplot of the number of reads from shared repeat clusters from the comparative analysis of the two species of *Tmesipteris*. The slope of the line is equal to the ratio of the GS of the two species (R = 2.01).

**Figure 3 ijms-24-02708-f003:**
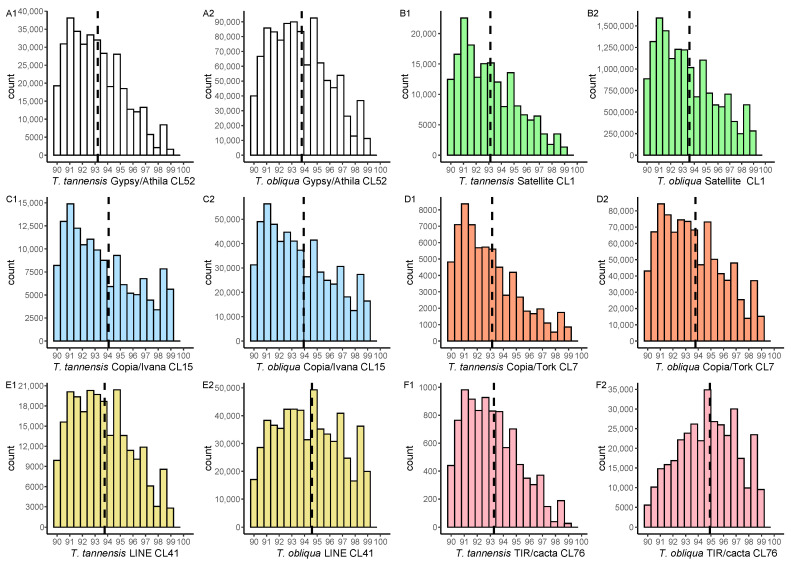
Intrafamily heterogeneity of repeats in *Tmesipteris*. (**A1**–**F2**) Histograms of percentage sequence similarity for read pairs in % (*x*-axis) from selected repeat families from *T. tannensis* and *T. obliqua*. The *y*-axis ‘count’ refers to the number of reads and dashed lines indicate the median heterogeneity values per cluster (CL).

**Figure 4 ijms-24-02708-f004:**
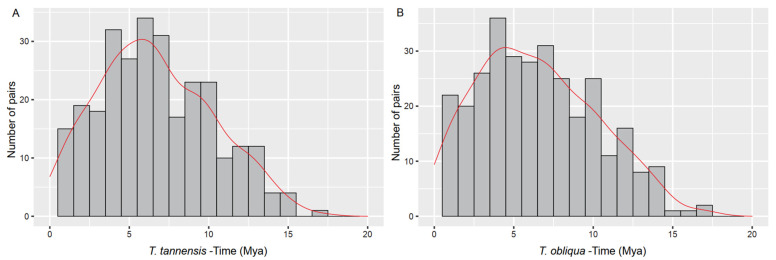
Average insertion times of LTRs of transposable elements in *Tmesipteris*: (**A**) *T. tannensis* and (**B**) *T. obliqua*.

**Table 1 ijms-24-02708-t001:** Repetitive DNA composition estimated in the individuals of *T. tannensis* and *T. obliqua* analyzed. Gbp = giga base pairs; TE = transposable element; rDNA = ribosomal DNA.

		Genome Proportion (GP)			
		*T. tannensis*		*T. obliqua*	
**Repeat Type**	**Lineage**	**[%]**	**[Gbp]**	**[%]**	**[Gbp]**
**Class 1 TE**					
**Ty1/Copia**		14.13	20.77	16.45	12.01
	SIRE	0.11	0.17	0.06	0.05
	Ale	1.45	2.13	1.40	1.02
	Ivana	5.00	7.34	4.51	3.29
	Tork	4.54	6.67	7.99	5.84
	Other	3.03	4.45	2.48	1.81
**Ty3/Gypsy**		9.36	13.76	9.07	6.62
	Tekay	0.26	0.38	0.10	0.07
	Athila	8.79	12.92	8.01	5.85
	Reina	0.00	0.00	0.04	0.03
	Other	0.32	0.46	0.93	0.68
**LTR-unclassified**		1.86	2.73	2.65	1.94
**Class 2 TE**					
	LINE	4.91	7.22	4.04	2.95
**DNA transposons**		3.39	4.99	4.27	3.12
	TIR/EnSpm-CACTA	1.62	2.39	1.65	1.21
	TIR/haT	1.68	2.47	2.62	1.91
	Helitron	0.09	0.13	0.00	0.00
**Other repeats**					
**Tandem repeats**					
	rDNA	0.23	0.17	0.27	0.20
	Satellite	6.47	9.78	8.04	5.87
**Unclassified**		14.61	21.48	15.49	11.31
**Small unclassified clusters (GP < 0.01%)**		14.31	21.03	16.97	12.39
**Total repeats**		69.27	101.83	77.26	56.40
**Single copy**		30.73	45.17	22.74	16.60

**Table 2 ijms-24-02708-t002:** Regression analysis statistics for different datasets. Regression between number of reads from; (i) repeat clusters classified Ty1/Copia elements, (ii) repeat clusters classified as Ty3/Gypsy elements, (iii) all clusters identified, and (iv) all clusters classified by RepeatExplorer2.

Repetitive Elements (*T. obliqua*/*T. tannensis*)	Slope	R^2^	*p*-Value
Ty1/Copia	2.131	0.914	<2 × 10^−16^
Ty3/Gypsy	1.785	0.943	<2 × 10^−16^
All	1.665	0.605	<2 × 10^−16^
All classified	1.904	0.886	<2 × 10^−16^

## Data Availability

The sequencing datasets presented in this study can be found in online repositories on the following link: https://www.ncbi.nlm.nih.gov/bioproject/PRJNA900075/ (accessed on 1 December 2022).

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
