# Peer review of "Giant Fern Genomes Show Complex Evolution Patterns: A Comparative Analysis in Two Species of Tmesipteris (Psilotaceae)"

_ijms, 2023, doi:10.3390/ijms24032708_

Round 1

Reviewer 1 Report

This is a short paper describing analysing repetitive DNA in two large fern genomes. RepeatExplorer is used to find the repeats and comparison are made. As fern genomes have not been analysed much in the past, I find this work interesting although looking at a wider range of species and more distant relatives would be more itneresting. I find that the MS as written does not work very well. The long and detailed M&M section at the end after the Discussion makes it very hard to understand what has been done when reading the Result section. I think some reorganisation is needed and explanations given in M&M need to be moved to  the Result sections.    

In more detail I comment as follows:

Part of the language needs to be tightened; it is in places too chatty and with awkward or misused expressions. I mention just a few from the Introduction, but the entire MS will need careful attention to language and style; e.g. they [the fern species] live mostly as epiphytes on tree fern species’,  ‘genomic gigantism to a level that has never been addressed before in any group’, ‘. Despite ongoing advances in genomics research, which make ambitious sequencing approaches more feasible than  ever before,’ ‘s the dearth of understanding on fern genomes’

I suggest to also cite the recent paper on  genome size in ferns by Schneider and co workers  (Fujiwara  et al 2023. Evolution of genome space occupation in ferns: linking genome diversity and species richness. Annals of Botany. In press. https://doi.org/10.1093/aob/mcab094

Abstract: I would mention that Paris japonica is an angiosperm  to indicate comparison of ferns and flowering plants.

Introduction should give a short overview about using raw read sequence analysis for repeat analysis – see also my comment to the beginning of the Result section.

The result section dives straight into describing the outcome of the analysis, but it is not clear for the reader what analysis are used. For example what is meant by cluster? This only makes sense if one reads the M&M section first or indeed the abstract to learn that RepeatExplorer was used and one is familiar with this kind of  analysis. Nothing of the power of raw read analysis and RepeatExplorer (albeit now used frequently for such analysis) is given in the introduction or here in the beginning of the Results.  

Table 1: what is the classification based on and how was it derived? Give more information in the heading including the database you used to classify the repeats.  Line 124 referring to Table 1 says “(see Table 1 for unique elements)” what does ‘unique’ mean; the main part of the sentence speaks of most repeat clusters containing reads from both species.

Line 137/38 “The fraction of the genome identified as single and low copy repeats (< 500 copies and not resolved by Repeatexplorer2) represents the ‘dark matter’ of the genome [27].”  I do not agree with the term ‘dark matter’. It is not used as term in ref 27 (Novak et al 2020); however I find this reference Bozgeyik 2022: The dark matter of the human genome and its role in human cancers.DOI: 10.1016/j.gene.2021.146084. Surely these transcribed ultra-conserved regions of non-coding RNAs is not what the authors want to refer to. And I, indeed, hope that most genes and regulatory sequences - not being repeats and hence part of this ‘unresolved fraction’ of RepeatExplorer - are extremely useful and essential and not a dark matter.  In a dedicated Discussion section you go more into detail of what you mean by ‘ dark matter’, and I agree that there might be left over degenerate by now low or single copy sequences no longer being part of a distinguished repeat class or cluster. Nevertheless, do I not agree that all single copy sequence or singlets as RepeatExplorer says are ‘useless’ dark matter. I suggest to modify your

Paragraph starting line 155: Be consistent of how you refer to clusters; you use e.g. cluster C76 and Cluster 18. Check elsewhere in the paper.

Insertion time: explain on what this analysis is based; e.g. the sequence similarity of the LTRs of complete elements.  

Figure 3: x-ais label is missing.  

I note a relatively large proportion of LINEs, unusual in angiosperms, can you comment.

Table S1: give source of DNA content and chr numbers

Reviewer 2 Report

The authors propose a manuscript titled “Giant Fern Genomes Show Complex Evolution Patterns: a Comparative Analysis in Two Species of Tmesipteris (Psilotaceae)”

The article is well structured and is original. In particular, this study takes into consideration a topic aspect on giant genomes are rare in the plant kingdom and were studied especilly on angiosperms and gymnosperms, while there are few genetic data available for ferns. This statement is true but must be referred also in other fields, for example there are less works in flora, vegetation, habitat, conservation on fern respect vascular plants (angiosperm and gymnosperm). This consideration must be evaluated in the introduction!!!

The authors discuss on genus Tmesipteris, that of mainly epiphytic ferns that occur in Oceania and several Pacific Islands. The authors declare correctly that only two species with giant genomes have been reported in the genus, T. tannensis and T. obliqua. Low-coverage genome skimming sequence data were generated in these two species and analysed using RepeatExplorer2 pipeline to identify and quantify the repetitive DNA composition of these genomes. The results show both species share a similar genomic composition, with high repeat diversity compared to taxa with small genomes. The study also highlights that, in general, characterised repetitive elements have relatively high heterogeneity scores, indicating ancient diverging evolutionary trajectories.

I appreciate the original idea of the work which with a few revisions will convince me and the editor to publish it on Journal.

Introduction

Weel structured and referenced but is necessary to introduce the reader in the correct way.

·    Rows 36-37. Please check as I already said and change the period in the suggested way: “Ferns are a group of plants of considerable phylogenetic interest [choose reference], which are less studied than vascular plants, as angiosperms and to a lesser extent gymnosperms for various scientific aspects, such as their taxonomy [e.g. Cárdenas et al. 2019], ecology, habitat and vegetation (e.g. Perrino et al. 2022), and the genome dynamics [e.g. Marchant et al. 2019]

Add reference

ü Cárdenas, G.G.; Lehtonen, S.; Tuomisto, H. Taxonomy and evolutionary history of the neotropical fern genus Salpichlaena (Blechnaceae). Blumea - Biodiversity, Evolution and Biogeography of Plants 201964, 1-22. https://doi.org/10.3767/blumea.2018.64.01.01

ü Perrino, E.V.; Tomaselli, V.; Wagensommer, R.P.; Silletti, G.N.; Esposito, A.; Stinca, A.Ophioglossum lusitanicum L.: New Records of Plant Community and 92/43/EEC Habitat in Italy. Agronomy 202212, 3188https://doi.org/10.3390/ agronomy12123188

ü Marchant, D.B.; Sessa, E.B.; Wolf, P.G. et al. The C-Fern (Ceratopteris richardii) genome: insights into plant genome evolution with the first partial homosporous fern genome assembly. Sci Rep 2019, 9, 18181. https://doi.org/10.1038/s41598-019-53968-8

·      Wel done. The sceintific name of plant species, as genera and species are reported correctly

Results and Material and Methods

Well done. Few suggestions

·      Table 1. Please add ine also under T. obliqua;

·      Table 1. I suggest do add a legend in order to have a complete picture of some acronyms as Gbp, Ty. Not all reader are genetist!!! In this way please check also material and methods chapter

Conclusion

Please two more words on the future research perpectives in this field.

References

Please check and format in the correct way, the number of references are duplicated.
